# Hierarchical Structured Polyimide–Silica Hybrid Nano/Microfiber Filters Welded by Solvent Vapor for Air Filtration

**DOI:** 10.3390/polym12112494

**Published:** 2020-10-27

**Authors:** Dawei Li, Ying Shen, Lanlan Wang, Feng Liu, Bingyao Deng, Qingsheng Liu

**Affiliations:** Key Laboratory of Eco-Textiles (Ministry of Education), Nonwoven Technology Laboratory, Jiangnan University, Wuxi 214122, China; ldw@jiangnan.edu.cn (D.L.); shenying1@sina.cn (Y.S.); 15852776701@163.com (L.W.); fengliu_windy@163.com (F.L.); qsliu@jiangnan.edu.cn (Q.L.)

**Keywords:** electrospinning, hierarchical structure, hybrid nano/microfiber filters, air filtration

## Abstract

Electrospun polymer membranes were considered to be promising materials for fine particulate matter (PM) filtration. However, the poor mechanical properties of the electrospun membrane restricted their application for pressure-driven air filtration. Herein, strength-enhanced electrospun polyimide (PI) membranes were demonstrated via a synergistic approach. Solvent-vapor treatment was utilized to introduce extra bonding at the cross points of PI nanofiber, while SiO_2_ nanoparticles (SiO_2_ NPs) were used to reinforce the body of nanofibers. The mechanical strength and filtration performance of hybrid membranes could be regulated by adjusting the quantity of SiO_2_ NPs. The tensile strength of the pure PI membrane was increased by 33% via adding 1.5% SiO_2_ NPs, which was further promoted by 70% after solvent-vapor treatment. With a slight reduction in pressure drop (6.5%), the filtration efficiency was not greatly suppressed by welding the SiO_2_ NP hybrid PI nanofibers. Moreover, the welded composite filter showed high particulate (0.3–1.0 μm) filtration efficiency (up to nearly 100%) and stable pressure drop throughout the 20 tested filtration cycles.

## 1. Introduction

With economic and industrial development, pollutants and their impact on human health have become a priority in recent research. The fine particulate matters (PM) are the primary abundant pollutant in the air, characterized by their small profile and their high specific area. These features allow them to be easily inhaled and passed into the bronchi and lungs through the respiratory tract and cause serious health problems [1]. Several studies have linked long-term exposure to PM to a variety of diseases, including strokes, lung disease, cardiovascular disease, and even cancer [2,3,4]. Therefore, effectively removing PM from polluted air is an urgent issue.

The porous membrane-type filter and fibrous membrane-type filter are the predominant, widely used conventional air filter until now [5,6,7,8,9]. The porous one made of the solid substrate has interconnected pores in the membrane, which can block the particles, while gas passes through the pores. However, the porosity of these filters is quite low (<30%), which means that high filtration efficiency is always accompanied by high filtration resistance. The fibrous membrane-type filters consisting of well-aligned or randomly arranged fibers form a reticular support structure and tortuous pore channels, making it attractive in various fields, which turned out to be a feasible and promising method to fight against air pollution. The porous structure and open pores allow the easy passage of air molecules, which will also block the particulate matters with higher capacity. Until now, various methods have been invented to prepare high-performance air filter materials, including melt-blowing [10,11,12], spun-bonding [13,14], and needle-punching [15,16]. These fibrous structures consisting of fibers of micro-scale could block the particles in the air through thick physical barriers along with adhesion. Nevertheless, the fiber-based filters prepared via these methods are suffering from high air resistance due to the direct strike of air molecules against the fibers and densely compacted structure of fiber assembly. Electret materials are also applied in the air filtration, as they can attract PM due to fortissimo electrostatic force. However, the low initial pressure drop is accompanied by safety hazards and a short service life [12,17,18].

The nanofiber, however, is known for its large surface area and high surface energies compared to the micrometer-sized fiber; consequently, the interactions with PM and the filtration efficiencies are impressively enhanced [19,20]. Moreover, around the periphery of fibers, the aerodynamic behavior of airflow was proved to be changed significantly when the size of fibers reduces to the nanoscale [21,22,23], which is called the “slip flow phenomenon”. It was believed that the single fiber would not cause any turbulence as the airflow passed by when the fiber diameter was close to 65 nm. Therefore, a significant reduction in drag force of the airstream could be achieved [21,24]. Electrospinning is found to be effective in capturing fine particles relying on their ultralow diameters, tunable porous structure, and high specific area [19,25,26]. Nevertheless, due to the dense packing of nanofibers (with a limited porosity of <90%), electrospinning nanofiber membranes (ENMs) tend to raise the pressure drop significantly.

In previous reports, the pressure drop is positively correlated with the solid volume fraction of the filter when this filter has a fixed diameter and thickness [21]. For an outstanding functionalization of filter membranes, hierarchical structures, especially the roughness in nanoscale on the surface of electrospun nanofibers, are exceptionally critical [27,28,29]. The formation of protuberance on the surface of nanofibers could significantly improve the surface roughness as well as increasing the effective surface area, leading to the increased project area of the nanofiber at the inlet. Therefore, it is hypothesized that the filtration performance of the membranes will be further improved by introducing nanoparticles on the surface of electrospun nanofibers. On the other hand, inorganic electrets could promote charge trapping, which could enhance the filtration properties [17]. Silica nanoparticles (SiO_2_ NPs) electrets are considered as promising alternatives for novel fibrous materials conception for air filtration, due to their outstanding electret behavior stems from the dipole orientation and the charge storage [30,31]. 

Although electrospun mats of hybrid polymer nanofibers could meet most of the demands in terms of filtering performance, the weak mechanical strength limits their further application. Thus, strength enhancement of electrospun filters is particularly critical, which can be typically classified into three main strategies: twisting electrospun nanofibers into yarns [32,33], reinforcing fibers by incorporating extra materials [34], and welding fibers with more contact points [35,36,37]. Changing the nonwoven mats into twisted nanofiber yarns could improve the tensile strength along the axis of yarn; however, the resulted fabric would suffer from reduced porosity and less anisotropy, which is less suitable in the application of filtration. Adding inorganic nanoparticles is another typical reinforcement method to boost the mechanical strength of ENMs. Other strategies, including heating and photo and chemical cross-linking at their cross points, were utilized to strengthen the ENMs. Regardless of the enhancement in mechanical properties, these methods are too complex to operate and typically lead to dimensional shrinkage caused by the relaxation of molecule chains in the amorphous region [38,39]. Whereas, a systematic study of the synergistic effect of inorganic particles reinforcement and extra welding on mechanical and air filtration performance of ENMs has not yet been reported.

In this contribution, electrospun polyimide (PI) nanofibers were selected as building blocks for their brilliant thermal and physicochemical stability as well as their high mechanical strength [40,41]. Firstly, hierarchically structured SiO_2_ NPs hybrid PI (SiO_2_@PI) fibrous filters with robust filtration performances were fabricated with the incorporation of SiO_2_ NPs through electrospinning. Subsequently, inter-fiber bonding was generated by positioning the as-spun nanofiber in the fume of solvents, which could further improve the mechanical strength. Notably, PI nanofibers were reinforced using as-received SiO_2_ NPs without modification. On the other hand, solvent-vapor treatment was easily conducted with negligible shrinkage caused in the dimension. Benefitting from this unique inorganic–organic composite structure, a new filter with enhanced mechanical strength and promoted high filtration efficiency with recyclability could be prepared. 

## 2. Experiments 

### 2.1. Materials

Copolyimide staple fibers (brand name: P84), which belong to micro denier-fibers, was supplied by Evonik (Althofen, Austria). Hydrophilic SiO_2_ NPs with a mean particle size of 12 nm were purchased from King Chemical (Shanghai, China). N, N-Dimethylacetamide (DMAc) and N, N-dimethylformamide (DMF) were obtained from Sinopharm Chemical Reagent (Shanghai, China). All the chemicals were used as received.

### 2.2. Methods

#### 2.2.1. Preparation of SiO_2_@PI Hybrid Membranes

P84 staple fibers purchased from Evonik were pretreated and rinsed with acetone and ethanol to remove impurities, followed by drying under vacuum at 60 °C for 2 h. PI solution was prepared by dissolving 1 g P84 stable fibers in 10 mL DMAc, then magnetic stirring for 2 h at 80 °C to obtain a homogenous PI solution 10% (*w/v*). SiO_2_@PI solution was further generated by adding SiO_2_ NPs into the PI solution with the ratios of 1.0, 1.5, 2.0, 3.0, 4.0% by weight to PI, respectively. To generate uniform dispersion, the PI solution was stirred at 300 rpm for 2 h at 80 °C, followed by ultrasonic treating for 1 h. The SiO_2_@PI nanofiber membranes were manufactured on the electrospinning apparatus under the process condition of a high voltage of 25 kV, collecting distance of 20 cm, and feeding rate of 1 mL/h. All experiments were carried out at room temperature and 50–60% room humidity. 

#### 2.2.2. Preparation of PPS–SiO_2_@PI Composite Filters

To test the filtration performance, the SiO_2_@PI hybrid nanofiber membrane was firstly deposited on the support layer of polyphenylene sulfide (PPS) needle-punched nonwoven, which was fixed on the rotating drum (80 rpm/min). Afterward, the composite filter was cut into a circle with a diameter of about 15 cm and placed in a closed vial with DMF vapor (generated from DMF solution) at 40 °C for 0.5 h. Thus, extra bonding would be generated between the PPS nonwoven and PI electrospun nanofiber membrane by partially dissolving PI nanofiber in the fume of DMF vapor. The composite filter with bonded hybrid nanofibers was dried at ambient temperature for 12 h to allow the evaporation of the residual solvent.

### 2.3. Characterizations

The surface and cross-sectional morphologies of SiO_2_@PI hybrid nanofiber were determined via the scanning electron microscope (SEM, Japan). The diameter of fibers was measured in the SEM images with the help of image processing software ImageJ. The chemical composition of SiO_2_@PI hybrid nanofiber was analyzed by Fourier transform infrared (FTIR) spectroscopy (Thermo Fisher Scientific, Waltham, MA, USA) in the range of 400–4000 cm^−1^. X-ray diffraction (XRD) spectra were recorded on an X-ray diffractometer (Bruker AXS GmbH, Germany) with Cu X-ray source and a scanning range from 5 to 45°. The pore size of samples was tested on capillary flow porometry (CFP-1100A, Skei, Ithaca, NY, USA). Water contact angles (WCA) (10 μL) were calculated using a contact angle goniometer (Rame Hart, Succasunna, NJ, USA) equipped with an HD camera.

The thermal shrinkage behavior of the SiO_2_@PI hybrid membranes was tested by calculating the specific value of the area before and after annealing at 220 °C for 1 h. Thermogravimetric (TG) analysis was scanned from 50 to 800 °C at a heating rate of 10 °C/min under N_2_ atmosphere using a Q500 (TA Instrument Company, Milford, MA, USA). Differential scanning calorimeter (DSC) measurement was carried out from 30 to 400 °C with a heating rate of 10 °C/min under N_2_ atmosphere (Q200, TA Instrument Company, Milford, MA, USA).

The tensile test was performed on the Shimadzu testing system (EZ-S, Shanghai Bahens Instrument Co., China). Rectangular samples with a length of 50 mm and a width of 10 mm were tested at a strain rate of 3 mm/min.

The filtration efficiency, the pressure drop, and the air permeability of the membranes were tested as described previously [42], using an automated filter tester (LZC-H, Suzhou Hua Da Instrument and Equipment Co., China). The experimental set-up (automated filter tester, Suzhou Hua Da Instrument and Equipment Co., China) is shown in Figure 1. The filtration efficiency (at a flow rate of 84 L/min) and air permeability were calculated using Equations (1) and (2), respectively.
(1)$$\eta =(1-\frac{C_{down}}{C_{up}})\times 100$$
where *η* is the filtration efficiency; *C_down_* and *C_up_* are the number concentration of particles at filter downstream and upstream, respectively.
(2)$$R=\frac{Q}{A}\times 167$$
where *R* is the air permeability of the measured materials; *Q* is the air flow rate under 200 Pa pressure drop; and *A* is the test area used in this test (50 cm^2^).

The quality factor (QF) was defined as follows:(3)$$QF=-\frac{\ln(1-\eta )}{\Delta P}$$
where *η* is the filtration efficiency of the filter; ∆*P* is the pressure drop of the airflow crossing filter.

## 3. Results and Discussion

### 3.1. Morphology of Hybrid Membranes

SiO_2_@PI hybrid nanofiber with uniform morphology was prepared via electrospinning. Herein, hydrophilic SiO_2_ NPs were utilized due to the uniform distribution in the spinning solution. The morphologies and size distribution of electrospun SiO_2_@PI hybrid nanofibers embedded with different contents of SiO_2_ NPs (0, 1.0, 1.5, 2.0, 3.0, and 4.0% by wt.) are shown in Figure 2. It could be observed that pure PI nanofibers exhibit smooth surface topology with an average diameter of 229.34 ± 55.50 nm. The participation of 1.0% SiO_2_ NPs led to rough nanofibers of a relatively smaller diameter (165.68 ± 81.76 nm). It could be found that the roughness of the nanofibers further increased and the average diameter decreased to 157.00 ± 72.40 nm when the blending ratio of SiO_2_ NPs increased to 1.5%. It was indicated that the inclusion of SiO_2_ NPs in PI nanofiber membranes could significantly reduce the fineness of the hybrid nanofibers, which was attributed to the weakened intermolecular Van der Waals’ force between the PI molecule chains due to the existence of SiO_2_ NPs. However, some SiO_2_ NPs began to agglomerate, and the shrinkage of fiber caused by solvent volatilization did not change dramatically during electrospinning with increasing SiO_2_ NP contents, generating large clusters on the surface, and the diameter of SiO_2_@PI nanofibers increased to 385.80 ± 169.39 nm when the SiO_2_ nanoparticle contents increased to 4.0%. 

Furthermore, a notable and progressive increase in the wrinkles and nano protrusions on the surface was observed with the increase in SiO_2_ NP contents, as shown in Figure 2, (a_1_–f_1_) and (a_2_–f_2_). Notably SiO_2_ NPs showed a uniform distribution on the surface of PI nanofibers, leaving outstanding humps and bulges on the surface. That could be awarded to phase separation induced by the fast evaporation of the solvent. The irregular surface of PI nanofibers could improve the surface roughness as well as the efficacious surface area, leading to a boost in the hydrophobicity of the SiO_2_@PI hybrid membranes. As illustrated in Figure 3c, the water contact angles (WCAs) of 1.0% SiO_2_@PI, 1.5% SiO_2_@PI, and 2.0% SiO_2_@PI were 122.5, 127.0, and 130.6°, respectively, indicating a rise in WCAs with the adding of SiO_2_ NPs. According to the Wenzel model [43,44], the rougher the surface was, the more hydrophobic the samples becomes. The formation of an interval distribution of SiO_2_ NPs and PI polymer could significantly increase the hydrophobicity of the composited nanofiber. However, further increasing the content of SiO_2_ NPs would result in a slight decrease in WCA, which could be ascribed to the coating of hydrophilic SiO_2_ NPs on the surface of PI nanofibers (Figure 2, e_1_–e_2_ and f_1_–f_2_).

Additionally, the Knudsen number could be calculated via the following formula: (4)$$K_{n}=\frac{2\lambda }{d_{f}}$$
where *λ* and *d_f_* is the mean free path of air molecules (*λ* = 65.3 nm) and fiber diameter, respectively. Accordingly, the flow regime of the gas can be divided into four different flow regimes. The free-molecular flow regime, transition flow regime, slip flow regime, and continuum flow regime (Figure 3a) [21]. Together with the fiber diameter of SiO_2_@PI hybrid membranes, the air flow around PI nanofibers belongs to the transition flow regime. The corresponding *K_n_* values of PI, 1.0% SiO_2_@PI, 1.5% SiO_2_@PI, 2.0% SiO_2_@PI, 3.0% SiO_2_@PI, and 4.0% SiO_2_@PI were 0.55, 0.78, 0.83, 0.66, 0.56, and 0.34, respectively. According to the slip flow phenomenon principle (as shown in Figure 3b), when the fiber diameter got close to 65.3 nm, the velocity of air flow was non-zero on the surface of a single fiber. Therefore, the drag force of the airstream could be reduced significantly. Thus, we could reasonably assume that the SiO_2_@PI hybrid membranes showed the features of the lip-effect, especially for 1.5% SiO_2_@PI hybrid membrane. 

### 3.2. Chemical Structure 

The as-prepared membranes were characterized by FTIR and XRD to further investigate their chemical compositions and structure. After adding SiO_2_ NPs, the main functional groups of PI at 1779, 1714, 1359, and 720 cm^−1^ is C=O asymmetric stretch, C=O symmetric stretch, C-N stretching variation, and C=O bending variation in the resulting imide structures, respectively, and they all remained (Figure 4a) [45]. In addition, the peak intensity of the O–H vibration band (3487–3682 cm^−1^) and C–H band (2822–3137 cm^−1^) decreased with increasing SiO_2_ NP contents. It was also notable that the characteristic peak of Si–O bonds at around 1095 cm^−1^ might be overlaid by the C–O bonds [46]. Additionally, it was obvious that the width of diffraction peak (around 10°) was firstly decreased when the contents of SiO_2_ NPs increased to 1.5% and then enhanced with the further increased SiO_2_ NP contents, as shown in Figure 4b. The results of this study were dovetailed with many experimental measurements [47,48,49]. The reason was that the proper contents of inorganic nanoparticles could form a tight adhesion between the NP and PI molecules. The increased width of diffraction peak was due to the inhibition of crystallization by the particle agglomeration of SiO_2_ NPs during the solidification process. 

### 3.3. Thermal Performance

Thermal stability is another crucial parameter for the safety characteristics estimation of the high-temperature filters. The thermal shrinkage behaviors of membranes were tested by heating at 220 °C for 1 h, as shown in Figure 5a. The PI membrane showed significant shrinking after thermal testing (about 13%). As for the nanofiber membrane incorporated with SiO_2_ NPs, only slight shrinkage could be tested for the 1.5% SiO_2_@PI hybrid membrane (about 2.5%), which showed a color change from white to light yellow. While 3.0% SiO_2_@PI hybrid membrane could maintain its original dimensions with less shrinkage, remaining its original color. 

As for the quantitative study of the thermal property, TG testing was also performed (shown in Figure 5b). As observed, all the composited samples showed significant enhancement in thermal stability in comparison with a pure PI membrane, which displayed a similar mass loss behavior along with the increase in heating temperature. In the low temperature range (about 160 °C), the minimal weight loss was caused by the removal of residual organic solvent. Subsequently, larger weight loss was detected at around 450 °C, which was mainly due to the self-crosslinking, cyclization, and dehydrogenation of the membranes. As the temperature increased, the exothermic peaks appeared at 630 °C (maximum degradation rate temperature, T_max_), which was ascribed to the continuous release of the oxidation of carbon and carbon monoxide. The T_max_ values of the SiO_2_@PI composites firstly increased to 636.41 °C for 2.0% SiO_2_@PI composites and then decreased to 626.46 °C for 4.0% SiO_2_@PI composites, which were all higher than that of the PI nanofiber. The enhancement of the thermal stability could be attributed to the introduction of thermal stable SiO_2_ NPs. Additionally, the interaction and physical interpenetration of SiO_2_ NPs were further strengthened on a nanometer or even molecular level after electrospinning into nanofibers. Thereby, the decomposition of polyimide might be delayed by the incorporation of SiO_2_ NPs. Additionally, it is noted that in the systems where the silica phase had no chemical bonding or other interactions with the polymer, only a small thermal stability improvement was observed. After heating up to 800 °C, the weight dropped to the lowest, manifesting the complete combustion and decomposition of PI molecules. However, in the present study, the residual weight after being heated at 800 °C increased with the rise in the SiO_2_ NP contents, which was much higher than the incorporated SiO_2_ NP percentage. These results also confirmed the successful incorporation of the SiO_2_ moiety in the hybrid membranes and its positive effects on the PI stability. The thermal behavior of hybrid membranes was also evaluated by DSC measurements (Figure 5c). The PI and SiO_2_@PI hybrid membranes showed no significant peaks from 30 °C to 400 °C, indicating excellent thermal stability in the medium and low temperature environment.

### 3.4. Mechanical Property

As shown in Figure 6a, it was obvious that the strength of the SiO_2_@PI hybrid membranes was promoted as the SiO_2_ content increased from 0 to 1.5 wt.%, and the SiO_2_@PI hybrid membranes with 1.5 wt.% SiO_2_ NPs performed the highest tensile strength (1216.4 kPa, 33% enhancement compared to the pure PI membrane) among all samples. It could be concluded that the addition of SiO_2_ NPs effectively improved the strength of PI nanofiber membrane. Since the inorganic nanoparticles (SiO_2_) incorporating inside PI nanofibers could play a crucial role in absorbing extra energy and dispersing strength during stretching process. However, when the loading of SiO_2_ NPs increased from 2.0 to 4.0 wt.%, the tensile strength dropped significantly by more than 68%, due to the SiO_2_ NP agglomeration, which broke the continuity of the PI molecule along the PI nanofiber, causing weak sections in the nanofiber structure. In addition, all SiO_2_@PI hybrid membranes possessed lower tensile strains than pristine PI membranes, which is attributed to the introduction of SiO_2_ NPs into PI nanofibers, leading to slightly less flexibility. As shown in Figure 6b, Young’s Modulus of SiO_2_@PI hybrid membranes increased firstly and then decreased sharply with increased SiO_2_ NP contents. It should be attributed to the polymer–filler interaction and reduced packing density (Figure 2).

### 3.5. Porous Structure

It is widely believed that porous structure has a crucial impact on the filtration performance of fibrous materials. Herein, to investigate the influence of SiO_2_ NP contents on the filtration performance of filters, the porous structures and pore size of all hybrid membranes were studied. As illustrated in Figure 7, the pore sizes of all SiO_2_@PI hybrid membranes were in the range of 1.70–5.06 μm, which were higher than that of pure PI membranes (1.37 μm), which was mainly due to the introduction of SiO_2_ NPs resulting in the reduction in the packing density of membrane. The results also suggested that the maximum and mean pore size firstly decreased and then rose with the increasing SiO_2_ NP contents. Pore size was positively correlated to the fiber diameter of relevant membranes (Figure 2). Meanwhile, the microstructure and interconnectivity of the pores could be characterized by the bubble point pressure, which firstly rose from 1.329 to 2.479 psi and then reduced to 0.329 psi with the increasing SiO_2_ NP contents. The increase in the bubble point pressure of relevant membranes could be owing to the smallest fiber diameter and more SiO_2_ NPs protruding from the surfaces of fibers, or the interspace among the fibers could significantly improve the complexity of the hybrid membrane pore structure. Additionally, when further increasing SiO_2_ NP content, the increasing maximum pore size causing the value of the bubble point pressure decreased. 

### 3.6. Filtration Performance

The filtration performance of charge neutralized dioctyl sebacate (DEHS) particles in 0.3, 0.5, and 1.0 μm (PM_0.3_, PM_0.5_, and PM_1.0_) for different needle-punched PPS composite filters were tested (Figure 8). Figure 8b presents the filtration efficiency and pressure drop of the pure needle-punched PPS filter (PPS), polytetrafluoroethylene (PTFE) emulsion impregnated PPS composite filter (PPS–PTFE), PI nanofiber membrane composite filter (PPS–PI), and SiO_2_@PI hybrid nanofiber membrane composite filter (PPS–SiO_2_@PI) at the airflow velocity of 84 L/min. It was visible that both the filtration efficiency and pressure drop of the PPS composite filters were higher than that of the pure PPS filter, especially for the PPS–PI and PPS–SiO_2_@PI filters. Meaningfully, the PPS–SiO_2_@PI filter showed the best filtration performance compared with the other filters (Figure 8c), indicating that SiO_2_ NPs could capture more charges due to the permanent dipole orientation character of SiO_2_ NPs. 

In corroboration of the superiority of PPS–SiO_2_@PI composite filters as filter media, the filtration performance of PPS–SiO_2_@PI composite filters was further optimized by doping different SiO_2_ NP content (PPS–PI, PPS–1.0%SiO_2_@PI, PPS–1.5%SiO_2_@PI, PPS–2.0%SiO_2_@PI, PPS–3.0%SiO_2_@PI, and PPS–4.0%SiO_2_@PI). As shown in Figure 8d and e, it could be observed that both filtration efficiency and air permeability of the PPS–SiO_2_@PI composite filters were higher than that of the PPS–PI filters. Compared with the PPS–PI filters, all PPS–SiO_2_@PI composite filters had a modest increase (14~23%) in filtration efficiency for PM_0.3_. We also found that the filtration efficiency of PM_0.3_ was more than 85% for the PPS–SiO_2_@PI composite filters, especially for PPS–1.5%SiO_2_@PI filters (94.378%), while that of PPS–PI was only 71.501%. However, the pressure drop illustrated a contrary trend by adding SiO_2_ NPs. The nano-protrusions formed by the inorganic nanoparticles could raise the projected frontal areas of nanofibers at the inlet and impose a pressure gradient due to the increased tortuosity, which allowed the penetration of air flow partly and intercepted PM effectively (Figure 8a). For the PPS–SiO_2_@PI composite filters, as mentioned above, the fiber diameter and mean pore size of the SiO_2_@PI hybrid membranes decreased and the intercellular structure became more complex as the SiO_2_ NP contents rose from 1.0 to 1.5%, contributing to the improvement of the filtration efficiency of PPS–SiO_2_@PI composite filters, especially for PM_0.3_. When the SiO_2_ NP contents were over 2.0%, the filtration efficiency of PPS–SiO_2_@PI composite filters reduced due to the increase in fiber diameter and pore size. To perform the overall evaluation of the filtration performances, quality factor (QF), a parameter that considers both filtration efficiency and pressure drop, were further calculated. Results showed that when the SiO_2_ NP contents increased from 0 to 1.5%, the QF value of the composite filters for PM_0.3_, PM_0.5_, and PM_1.0_ improved. However, when the SiO_2_ NP contents continued to increase to 4.0%, the QF value reduced (Figure 8f). Notably, compared with the other PPS–SiO_2_@PI composite filters, the PPS–1.5%SiO_2_@PI filters enjoyed a relatively higher QF value of 0.029 for PM_1.0_, indicating outstanding filtration performances for particles with a small diameter and great potential application in polluted air purification. Further investigation of the PPS–SiO_2_@PI composite filters was conducted on the membrane with 1.5% SiO_2_ NPs as the optimal condition.

### 3.7. Filtration Efficiency and Reusability of Welded Composite Filters

Results indicated that the filtration efficiency of welded PPS–1.5%SiO_2_@PI filters for all particles exceeded 90%, slightly smaller than that of the unwelded PPS–1.5%SiO_2_@PI filters (Figure 9a). It could be attributed to the slight increase in the average pore size from 1.70 to 1.77 μm after welding (Table 1). The changes in pore structure were evaluated by comparing the pore size of the composite filters before and after welding (Figure 9b). The number of the smaller pores (~1.68–2.00 μm) was slightly reduced when the filter was welded, corresponding to the increase in the overall mean pore size. In addition, Figure 9b indicated that large pores (3.5–4.5 μm) could be created as a result of solvent-vapor treatment. This was probably due to the fact that, during the welding process, PI nanofibers would be softened, and those with a smaller diameter were dragged to the adhesive onto the bigger ones, which would drag the fibers closer to each other and leave more local free volumes or bigger pores (as shown in Figure 9c,d). Moreover, the reduced bubble point pressure from 2.479 μm to 0.930 μm after welding was also supporting the change in pore structure. In addition, the synergistic effects of vapor-welding and SiO_2_ NPs joining on the strength enhancement were studied. An increase in tensile strength (from 1216.40 kPa to 1547.49 kPa) of the welded 1.5%SiO_2_@PI membrane could be confirmed (Figure 9e). The percentage of enhancement in the strength was 27%, compared to that of the unwelded 1.5%SiO_2_@PI membrane. The improved mechanical strength could be attributed to the inter-bonding of fibers at the cross points in-plane as well as in vertical-to-plane directions, which restricted mat delamination. Moreover, it was found that the tensile strength enjoyed a 70% enhancement, compared to the pure PI nanofiber membrane (Figure 6a). The improvement in mechanical strength by the combination of welding and SiO_2_ NPs reinforcement was more pronounced than that of a single strategy. More importantly, although the elongation of the PI membrane reduced after adding SiO_2_ NPs, the elongation of the SiO_2_@PI membrane could be compensated by solvent-vapor treatment in this work.

Reusability of welded PPS–1.5%SiO_2_@PI filters was determined by conducting a 20-cycle filtration for various particle sizes (PM_0.3_, PM_0.5_, and PM_1.0_), as shown in Figure 9f. Filtration efficiency for PM_1.0_ remained higher than 99% during the 20 filtration testing cycles. In the case of the PM_0.3_, the efficiency >95% could be obtained after nine filtration cycles. It was suggested that high-performance filtration of the welded PPS–1.5%SiO_2_@PI filters could maintain for at least 20 cycles. It should be noticed that the increase in filtration efficiency for PM_0.3_ was attributed to the particles that strongly attached to the membrane surface with increasing filtration cycles. The accumulation of particles in the filters would block more particles trying flow through the pores, thereby filtration efficiency was increased. Most notably, the pressure drops of welded PPS–1.5%SiO_2_@PI filters had almost no change for 20 testing cycles, which could be explained by the fact that most particles were attached to the fiber surface due to the electrostatic interaction but not tightly accumulated in the pores of the filters. The robustness of the filter in long-term usage was attested. In addition, the hybrid filter prepared in this study possessed comparable or higher filtration efficiency to other electrospun inorganic NP-filled polymer hybrid filters [27,50,51,52], as shown in Figure 10.

## 4. Conclusions

In this study, we have brought out an approach to improve the mechanical strength of PI nanofibrous membranes via a combination of solvent-vapor-induced welding and SiO_2_ NP reinforcement. The results showed that the SiO_2_ NPs distributed in the surfaces of PI nanofibers could generate hierarchical structures, which not only enhanced the mechanical properties but also promoted the filtration performance of composite filters. More importantly, the mechanical properties and filtration performance could be further tuned by adjusting the SiO_2_ NP content, and the welded 1.5%SiO_2_@PI showed the most balanced property (the increase in tensile strength and filtration efficiency by 70 and 20%, respectively). The reusability of the welded PPS–1.5%SiO_2_@PI composite filter for filtration of PM_0.3_, PM_0.5_ and PM_1.0_ was confirmed, as the filtration efficiency was maintained at at least 90% and up to nearly 100% without increasing the pressure drop after the 20 filtration testing cycles. Therefore, it is promising to employ this new route for the preparation of mechanically strengthened electrospun materials with high-performance air filtration.

## Figures and Tables

**Figure 1 polymers-12-02494-f001:**
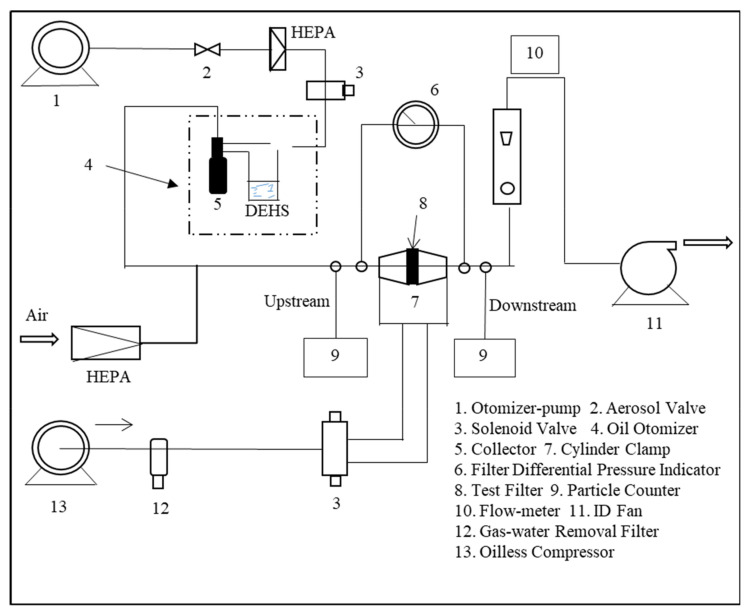
Experimental set-up for testing filtration efficiency.

**Figure 2 polymers-12-02494-f002:**
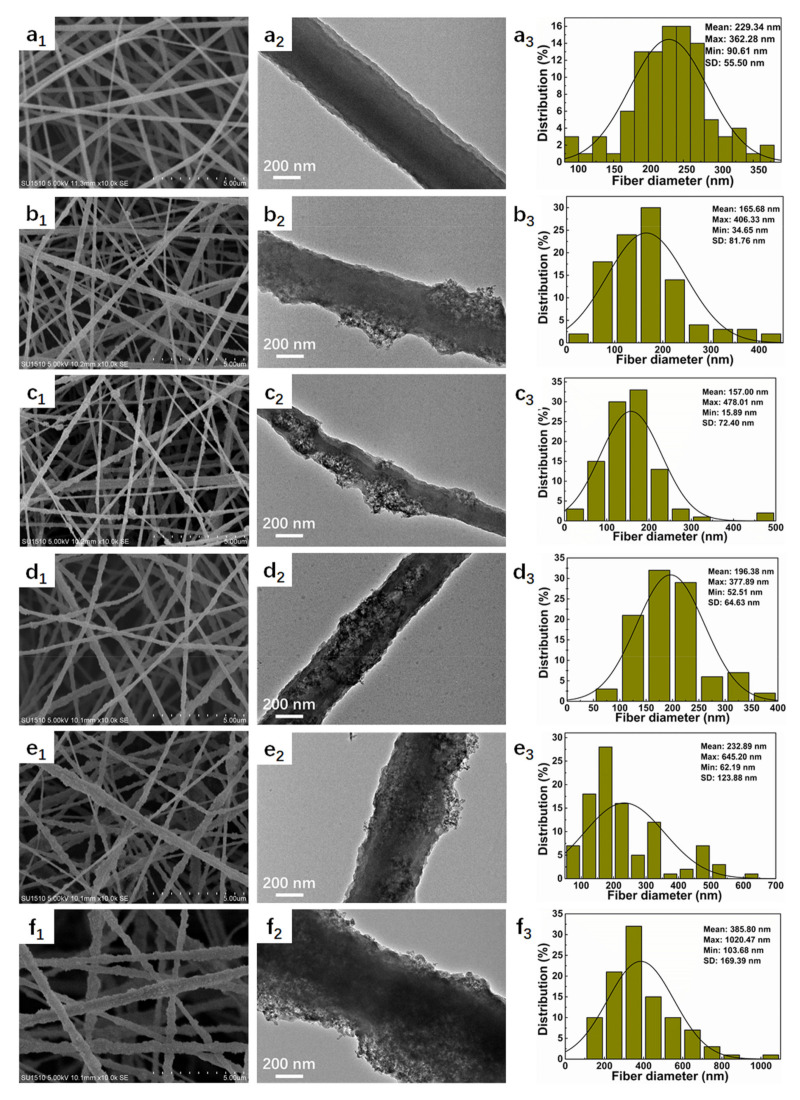
SEM images (**a_1_**–**f_1_**), TEM images (**a_2_**–**f_2_**), and fiber diameter distribution (**a_3_**–**f_3_**) of SiO_2_@ polyimide (PI) hybrid membranes with different SiO_2_ NP contents: (**a_1_**–**a_3_**) 0%, (**b_1_**–**b_3_**) 1.0%, (**c_1_**–**c_3_**) 1.5%, (**d_1_**–**d_3_**) 2.0%, (**e_1_**–**e_3_**) 3.0%, and (**f_1_**–**f_3_**) 4.0%.

**Figure 3 polymers-12-02494-f003:**
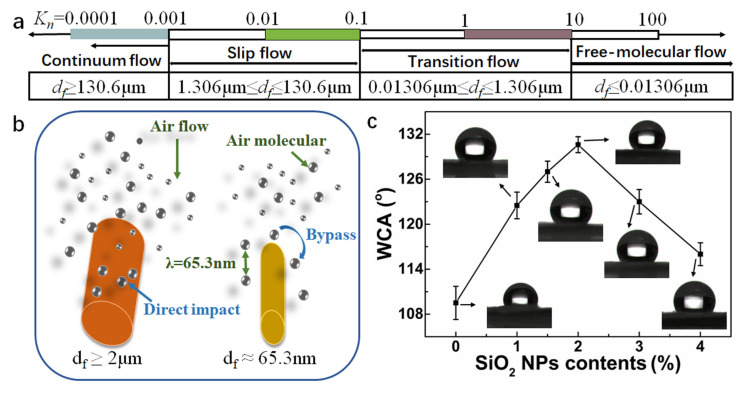
(**a**) The schematic representation of the scale bars of flow regime. (**b**) Schematic showing the mechanism of slip flow. (**c**) WCAs and the selected optical profiles of water droplets on various SiO_2_@PI hybrid membranes.

**Figure 4 polymers-12-02494-f004:**
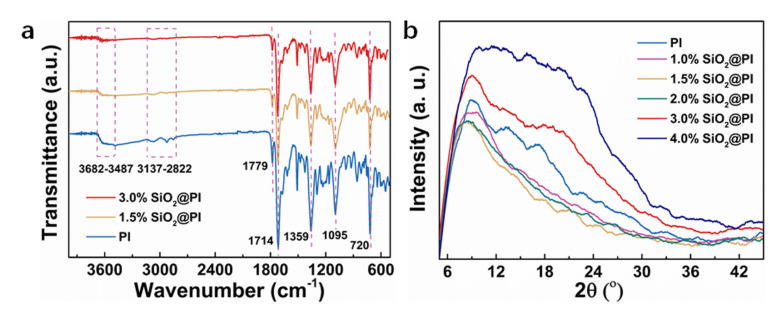
(**a**) FTIR spectra and (**b**) XRD patterns of PI and SiO_2_@PI hybrid membranes.

**Figure 5 polymers-12-02494-f005:**
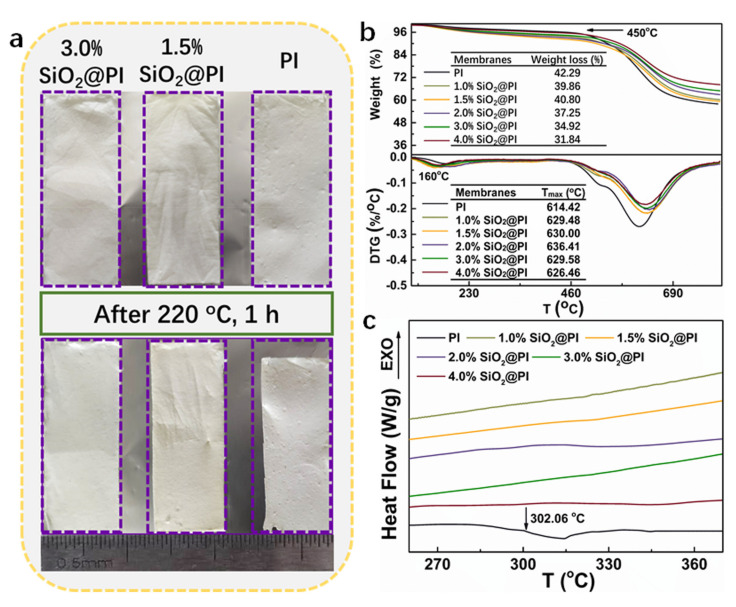
(**a**) Photos of PI and SiO_2_@PI nanofiber membranes before and after being exposed at 220 °C for 1 h; TG and the differential thermogravimetric (DTG) curve (**b**) and differential scanning calorimeter (DSC) curves (**c**) of the electrospun hybrid membranes with different SiO_2_ NP contents.

**Figure 6 polymers-12-02494-f006:**
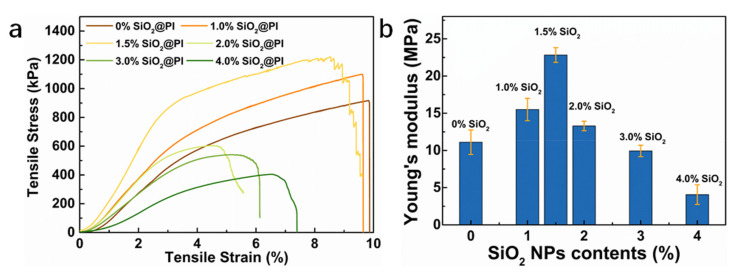
Mechanical characterization of SiO_2_@PI hybrid membranes: (**a**) tensile stress–strain curves; (**b**) Young’s Modulus.

**Figure 7 polymers-12-02494-f007:**
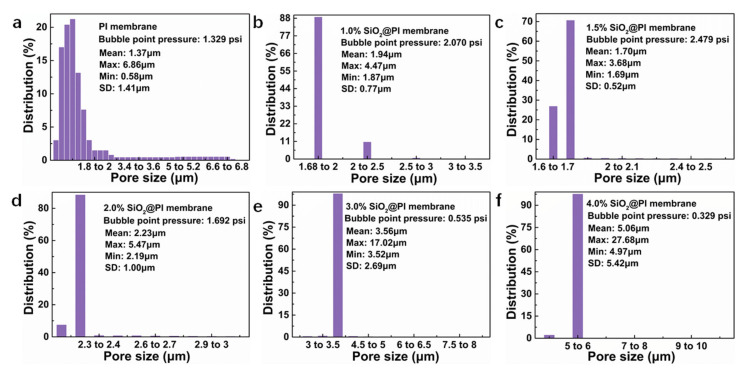
Pore sizes and their distributions of various SiO_2_@PI hybrid membranes. (**a**) PI, (**b**) 1.0% SiO_2_@PI, (**c**) 1.5% SiO_2_@PI, (**d**) 2.0% SiO_2_@PI, (**e**) 3.0% SiO_2_@PI, and (**f**) 4.0% SiO_2_@PI.

**Figure 8 polymers-12-02494-f008:**
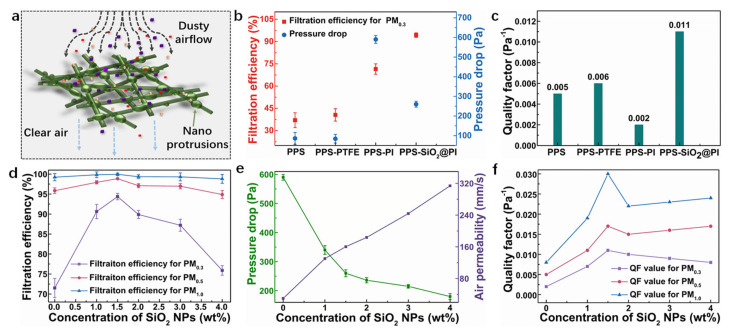
(**a**) The schematic diagram of air filtration through polyphenylene sulfide (PPS)–SiO_2_@PI composite filters; filtration performance (**b**) and quality factor (QF) values (**c**) of different kinds of PPS filters; filtration efficiency (**d**), pressure drop and air permeability (**e**), and QF values (**f**) of relevant PPS–SiO_2_@PI composite filters for PM_0.3_, PM_0.5_, and PM_1.0_.

**Figure 9 polymers-12-02494-f009:**
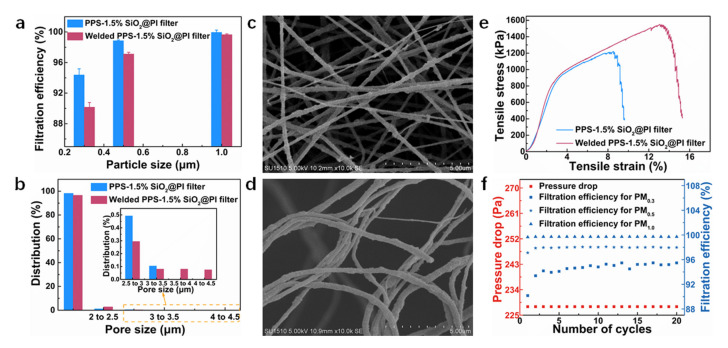
Filtration efficiency (**a**), pore size and its distribution (**b**), and tensile stress–strain curves (**e**) before and after welded PPS–1.5%SiO_2_@PI filters; SEM images of PPS–1.5%SiO_2_@PI filters before (**c**) and after (**d**) being welded by solvent vapor; (**f**) filtration efficiency of welded PPS–1.5%SiO_2_@PI filters in 20 cycles.

**Figure 10 polymers-12-02494-f010:**
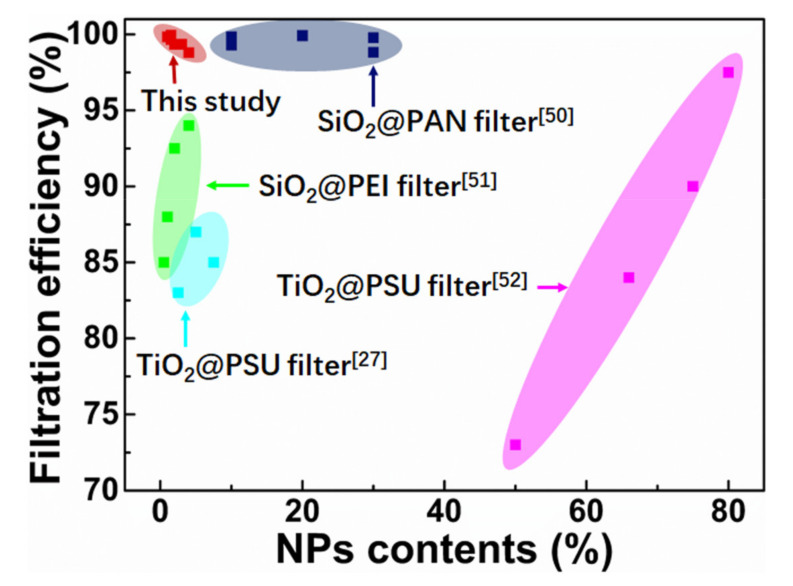
Ashby plot of NP contents versus filtration efficiency for other inorganic NP-filled polymer hybrid filters.

**Table 1 polymers-12-02494-t001:** Characteristics of PPS–1.5%SiO_2_@PI filters before and after being welded by solvent vapor.

Filters	Mean Pore Size (μm)	Maximum Pore Size (μm)	Bubble Point Pressure (psi)	Pressure Drop (Pa)
PPS–1.5%SiO_2_@PI	1.70	3.68	2.479	260
Welded PPS–1.5%SiO_2_@PI	1.77	7.98	0.930	243

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
