# Peer review of "Hierarchical Structured Polyimide–Silica Hybrid Nano/Microfiber Filters Welded by Solvent Vapor for Air Filtration"

_polymers, 2020, doi:10.3390/polym12112494_

Round 1
Reviewer 1 Report
The authors have significantly revised and improved the quality of the manuscript.
However, for a better understanding of the article, it is nessesary to make the following additions:
- P84 is a copolyimide, it is should be mentioning in the text;
- section "2.2.1. Preparation of SiO2 @ PI Hybrid Membranes" - probably the formation of membranes in a chamber with a humidity of 50-60% was carried out to create a porous structure of the membranes. If so, then it is necessary to indicate it in the text;
- section "2.2.2. Preparation of PPS-SiO2 @ PI Composite Filters" - why Composite Filters was placed in a closed vial with exactly DMF vapor;
- for a more detailed explanation of the study of morphology, thermal and mechanical properties, it is necessary to assume how the interaction between the particles SiO2 and copolyimide P84 occurs. Why does the introduction of a small amount of up to 2% hydrophilic nanoparticles lead to a significant hydrophobization of the surface compared to pure P84 (this fact seems to be the strangest), an increase in Tg and tensile strength;
- a separate question on contact angles - the SEM images show that the nanoparticles are not evenly distributed over the surface of the hollow fiber when a small amount of nanoparticles is introduced (somewhere on the surface there is a structure of pure P84). This means that when the contact angles were measured on different surface of the fiber,
different values were obtained. How many times have the experiments been carried out, and what was the measurement error. I would recommend re-measuring contane angles, because the data looks questionable.
Reviewer 2 Report
The authors have made a good effort to revise their manuscript and they have addressed my concerns listed in the previous round of reviews.
Author Response
Thanks for your comments.
This manuscript is a resubmission of an earlier submission. The following is a list of the peer review reports and author responses from that submission.
Round 1
Reviewer 1 Report
First of all, the very low quality of the manuscript presentation should be noted. There are a lot of misprints and unfinished sentences in the text, and in this form the manuscript cannot be considered for publication in the journal.
In addition, there are a number of questions about the results:
1. Lines 175-176. It is not very clear how can surface morphology of the membranes affect their hydrophobicity.
2. Misprint in the title of Fig. 2 (4.0 %).
3. Fig. 7a. It is kind of weird that total pore size distribution is more than 100 %.
4. Fig 8d. Why is filtration efficiency of the membranes higher for smaller particles (PM0.3, PM0.5) than for PM1.0.? At the same time, in Fig. 9a there is a reverse dependence.
5. Misprints in the title of Fig. 9 (a, b, c, d)
Reviewer 2 Report
The paper focuses on hierarchically structured polyimide (PI)-silica electrospun membranes and their potential application for air filtration. The membranes have been characterised by XRD, SEM, water contact angle (WCA), FTIR, DSC, TGA, capillary flow porometry, tensile properties and filtration efficiency.
In principle, the topic should be of interest for the readers of the journal. The manuscript is well-organised. The novelty of the work relates to the introduction of silica nanoparticles as reinforcing agents and the use of solvent vapor treatment to generate additional cross-links points to the PI membrane. It is unclear if those treatments are important modifications compared to previous work.
My suggestions are shown below:
- The novelty of the work should be articulated in much more detail.
- How the authors have quantified the enhanced cross-link density of the membranes following the solvent vapor treatment?
- Why silica NPs were selected rather than any other type of nanoparticles?
- The silica NPs used here are hydrophilic or hydrophobic? How this behaviour relates to the reported WCAs of the composites?
- The XRD spectrum of the PI should be compared with similar spectra reported in literature.
- The authors should compare the filtration efficacy of their materials with that reported for similar systems.
- What are the advantages and limitations of the materials presented here?
- TEM images to explore the distribution of silica NPs would be beneficial.
- The water contact angles reported are average values?
- For the WCA experiments what was the temperature of the water?
- The tensile properties reported are average values?
- What are the error bars for those measurements?